# Effects of Tillage and Cover Crops on Total Carbon and Nitrogen Stocks and Particle-Size Fractions of Soil Organic Matter under Onion Crop

Ana Carla Kuneski [1], Arcângelo Loss [2,*], Barbara Santos Ventura [2], Thiago Stacowski dos Santos [2], Lucas Dupont Giumbelli [2], Andria Paula Lima [3], Marisa de Cássia Piccolo [4], José Luiz Rodrigues Torres [5], Gustavo Brunetto [6], Claudinei Kurtz [7], Cledimar Rogério Lourenzi [2] and Jucinei José Comin [2]

1 Department of Soils and Natural Resources, Santa Catarina State University, Conta Dinheiro, Lages 88520-000, Santa Catarina, Brazil; mnaxica@gmail.com
2 Center of Agricultural Sciences, Federal University of Santa Catarina, Itacorubi, Florianópolis 88034-000, Santa Catarina, Brazil; bazinhasv@hotmail.com (B.S.V.); thiagoskisantos@hotmail.com (T.S.d.S.); lukdg@hotmail.com (L.D.G.); lourenzi.c.r@ufsc.br (C.R.L.); j.comin@ufsc.br (J.J.C.)
3 Soil Department, Federal University of Rio Grande do Sul, Agronomia, Porto Alegre 91540-000, Rio Grande do Sul, Brazil; andriapaulalima2@hotmail.com
4 Center of Nuclear Energy in Agriculture, University of São Paulo, Piracicaba 13416-970, São Paulo, Brazil; mpiccolo@cena.usp.br
5 Federal Institute of Triangulo Mineiro, Uberaba Campus, 4000 São João Batista Ribeiro St., Uberaba 38064-790, Minas Gerais, Brazil; jlrtorres@iftm.edu.br
6 Center of Rural Sciences, Federal University of Santa Maria, Camobi, Santa Maria 97105-900, Rio Grande do Sul, Brazil; brunetto.gustavo@gmail.com
7 Research and Agricultural Extension Company of Santa Catarina, Lageado Águas Negras, Ituporanga 88400-000, Santa Catarina, Brazil; kurtz@epagri.sc.gov.br
* Correspondence: arcangelo.loss@ufsc.br

**Abstract:** Onion (*Allium cepa* L.) is a food crop of economic importance. In Brazil, the crop typically occurs in a conventional tillage system (CT), which favors the mineralization and decomposition of soil organic matter (SOM) and the loss of carbon (C) and nitrogen (N). On the other hand, the no-tillage vegetable system (NTVS) operates based on sustainable pillars and bypasses the adverse effects of CT. This study evaluated the total C and N stocks and particle-size fractions of SOM in NTVS with single and intercropped cover plants compared to vegetable crops under CT. The NTVS evaluated treatments were as follows: (1) spontaneous vegetation (SV); (2) black oats (BO); (3) rye (RY); (4) oilseed radish (OR); (5) RY + OR; and (6) BO + OR. A treatment under CT with millet cover, a no-tillage system with (NTS) millet + velvet + sunflower cover, and a forested area as the original condition was also evaluated. Soil samples were collected in 0–5, 5–10, and 10–30 cm layers. Stocks of total organic C (TOC), total N (TN), particulate OC (POC), particulate N (PN), mineral-associated OC (MAOC), and mineral-associated N (MN) were evaluated. The highest stocks of TOC, TN, POC, and NM were found in NTVS compared to CT, and RY + OR showed the best results. The NTVS showed higher TOC and TN stocks with grasses and cruciferous intercropped than NTVS with SV and CT. POC and PN stocks increased in areas with single and intercropped RY and OR treatments. MAOC and MN stocks were higher than forest in RY + OR intercrop in the topsoil layer. RY and OR intercrop efficiently added C and N to the soil under NTVS. The consortium of millet + velvet + sunflower in NTS showed higher TOC, TN, POC, and PN stocks compared to the other treatments (0–5 and 0–30 cm). In general, the intercrop of cover plants is ideal for obtaining the NTVS maximum potential, favoring several mechanisms between soil, plant, and atmosphere, resulting in improved soil quality, increased organic matter, and higher stocks of C and N.

**Keywords:** no-tillage system; conventional tillage; particulate organic carbon; particulate nitrogen; cover plants intercrop; pear millet

## 1. Introduction

Onion (*Allium cepa* L.) is among the vegetables with the most significant production volume worldwide. There are 55,266 hectares of crops in Brazil and 1,563,986 tons of onions, the third most relevant oleracea, just after tomatoes and potatoes [1]. Santa Catarina (SC) is the largest national producer among the states, and Ituporanga is the largest state producer, known as the Onion National Capital, responsible for 23% of production [2].

Onion crops typically occur in a conventional tillage system (CT), in which periodic soil mobilization is performed by machinery and agricultural implements, such as disk plow, subsoiler, heavy offset disc harrow, and the rotary tiller. This management promotes crop residues fragmentation and its incorporation into the soil, increasing the exposed soil surface and, consequently, the soil temperature and moisture oscillations, causing rapid decomposition and mineralization of soil organic matter (SOM) [3,4]. The CT also causes changes in soil structure, such as the breakdown of aggregates, increasing the losses of carbon (C) and nitrogen (N) stocks that were previously physically protected inside the aggregates [5,6]. In soils with no plant cover, poorly structured and powdered by CT, there is an increase in erosive processes, which, combined with the massive use of pesticides and soluble fertilizers, increases the negative impacts on the environment and the production costs, decreasing the quality of produced onion bulbs [7,8].

The no-tillage vegetable system (NTVS) solves CT-adverse effects. The NTVS recommends restricting soil mobilization, continuous live or dead cover crops, and crop rotations [9]. This management system allows reduction and/or elimination of erosive processes; maintenance of the soil microorganisms balance; a decrease in the spontaneous vegetation population; an increase in soil aggregation; a reduction of temperature; a humidity fluctuation on the soil surface, favoring soil aggregation; and an addition of C, N stocks, and other nutrients [10,11].

Strategically, the NTVS, in the technical-scientific field, promotes plant health and, in the political–pedagogical area, a dialectical methodological conception [12]. The core principle of plant health promotion consists of techniques that promote plant comfort by minimizing stress with nutritional balance, water availability and pH [13,14]; improving the arrangement of soil particles, which promotes an increase in total porosity and greater root growth [15]; crop rotations, green fertilizers, and soil mobilization restricted to planting row [16]; decrease and search for the elimination of the use of soluble fertilizers and pesticides; and reduction of production and environmental costs while maintaining yield index [17].

Live plants or their crop residues produce soil cover. This plant material depends on the botanical and agronomic characteristics of the species and the environmental and management conditions. Several species of cover plants are recommended for this purpose and their intercrop recommendations [18]. The use of single species is the most traditional system, and intercrops that use two or more species aim to complement and enhance their beneficial effects on the soil and the environment [19]. The plants' intercrop with different C/N ratios benefits the soil when their residues are deposited on the surface since species with low C/N ratios, such as Fabaceae, quickly mineralize, and provide nutrients for subsequent plants. Plants with high C/N ratios, such as Poaceae, remain in the soil longer [20]. In addition, the use of plants with different plant residues decomposition rates, such as forage turnip + rye, increases the total organic carbon (TOC). Turnip + oats intercrop increases the particulate organic carbon (POC) compared to single turnip [21].

SOM plays a vital role in maintaining microorganisms, formation of soil aggregates, water storage, and nutrient supply to plants, particularly N [22–24]. SOM consists of several compartments, differing in their susceptibility to microbial decomposition. Thus, it is necessary to fractionate the soil organic matter to evaluate these compartments. Through the particle-size physical fractioning of soil organic matter, we can obtain two fractions with different labilities: the SOM particulate fraction, which is more labile and known as particulate organic matter (POM); and the persistence fraction, known as mineral organic matter (MOM) [25].

In POM and MOM, we can quantify the C and N contents. We call them particulate organic carbon (POC) and particulate nitrogen (PN) when we quantify these elements in POM, and carbon or nitrogen associated to clay + silt minerals (MAOC or MN) when we quantify the C and N of MOM [6,26,27]. The POM includes biomass, live organic matter, small plant debris fragments, among other non-humic substances. This compartment provides much of the easily accessible food for soil microorganisms and much of the readily mineralizable N. It can be easily increased by the addition of plant residues, as well as easily lost by soil management with periodic mobilization [28]. Interactions between minerals and organic colloids chemically protect MOM. It has high resistance to microbial attack due to its association with minerals surface and its localization inside the aggregates, inaccessible to microorganisms [29]. The MOM fraction comprises a complex structure with high reactivity and high molecular weight. Its cycling is slower, remaining long in the soil, acting on the soil aggregates stabilization and as a reservoir of nutrients [28].

These SOM fractions are important storers of C and N in the soil. They are continuously being renewed, decomposed, and renewed through new additions of organic material to the soil. The no-tillage vegetable system recommends adding organic material, protecting organic matter and soil aggregates, slowly and gradually decomposing organic waste, increasing the C and N stock in the soil, and reducing gases emissions into the atmosphere [30,31]. However, the conventional tillage system provides the loss of C and N stocks and other elements by using plowing and tillage practices, which accelerate the soil's organic waste decomposition and mineralization processes [32–34].

Therefore, it is necessary to develop studies that evaluate these SOM dynamics in the no-tillage vegetable system and the conventional tillage system with long-term vegetable crops. Further, the best plant cover for vegetables under the no-tillage system should be test. In this study over 10 years of cultivation in the no-tillage vegetable system were evaluated. Thus, we hypothesize that the intercrop of different plant species covering plants used for straw production in the no-tillage vegetable system increases the C and N stocks of the SOM particle-size fractions compared to using single plant species. The present study aimed to evaluate the soil's total carbon and nitrogen stocks and the particle-size fractions of the SOM in the no-tillage vegetable system with single and intercropped cover plants, comparing them with the vegetable crops under the conventional tillage system.

## 2. Materials and Methods

### 2.1. Location and Experimental Design

The experiment was implemented in 2009 at the Santa Catarina Rural Extension and Agricultural Research Enterprise (EPAGRI), located in the municipality of Ituporanga, Alto Vale do Itajaí region, State of Santa Catarina, Brazil (Latitude 27°24′52″ S, Longitude 49°36′9″ W and altitude of 475 m). According to the Köppen classification, the region's climate is humid mesothermal subtropical (Cfa), with an average annual temperature of 18 °C, average yearly precipitation of 1400 mm, and hot summers and infrequent frosts. The soil was classified as Cambissolo Húmico Distrófico [35] or Typic Hapluldult [36]. The area used for the experiment had a history of twenty years of onion crop under the conventional tillage system (plowing and grading) until 1996. As of 2009, vegetables were cultivated in a no-tillage system with an onion experiment. The spontaneous vegetation, according to Souza et al. [37], was desiccated in April 2009 using glyphosate herbicide when the experiment was set up. From then on, pesticides and mineral fertilizers were no longer used. At the experiment set up in 2009, in the 0–10 cm layer, the soil had the following physicochemical parameters: clay loam texture with 380 g kg$^{-1}$ of clay, 420 g kg$^{-1}$ of sandy, 320 g kg$^{-1}$ of silt, 23.20 g kg$^{-1}$ of total organic carbon (TOC), 39.65 of organic matter (OM), 1.8 g kg$^{-1}$ of N, 6.2 pH in water, 26.6 mg dm$^{-3}$ available P, 145.2 mg dm$^{-3}$ of exchangeable K (extracted by Mehlich-1), exchangeable Al 0.0 cmolc dm$^{-3}$, exchangeable Ca 7.2 cmolc dm$^{-3}$, exchangeable Mg 3.4 cmolc dm$^{-3}$ (extracted by KCl 1 mol L$^{-1}$), cation exchange capacity (CEC) 14.3 cmolc dm$^{-3}$, and CECpH 7.0 base saturation 76%. These analyses were made according to Tedesco et al. [38].

The implemented treatments in the no-tillage vegetable system consisted of different cover plants, single (cover crop) or intercropped, as presented in Table 1.

**Table 1.** Name of the intercrop or cover crop and sowing and harvesting (disposal) time for each treatment.

| Treatments | Intercrop or Cover Crop | Sowing Time | | Mowing or Desiccation Time |
| --- | --- | --- | --- | --- |
| | | Winter Species | Summer Species | |
| NTVS: spontaneous vegetation (SV) | Control: SV with the predominance of the species *Rumex obtusifolius*, *Amaranthus lividus*, *Cyperus* spp. And *Oxalis* spp. | Early autumn (April) | December: velvet bean | At flowering, cover was bedded with a cutting roller |
| NTVS: rye (RY) | Cover crop: RY (*Secale cereale* L.) with 120 kg ha$^{-1}$ of seeds | Early autumn (April) | December: velvet bean | At flowering, cover was bedded with a cutting roller |
| NTVS: black oats (BO) | Cover crop: BO (*Avena strigosa* Schreb) with 120 kg ha$^{-1}$ of seeds | Early autumn (April) | December: velvet bean | At flowering, cover was bedded with a cutting roller |
| NTVS: Oilseed radish (OR) | Cover crop: OR (*Raphanus sativus* L.) with 20 kg ha$^{-1}$ of seeds | Early autumn (April) | December: velvet bean | At flowering, cover was bedded with a cutting roller |
| NTVS: BO + OR | Intercrop: BO with 60 kg ha$^{-1}$ of seeds + OR with 10 kg ha$^{-1}$ of seeds | Early autumn (April) | December: velvet bean | At flowering, cover was bedded with a cutting roller |
| NTVS: RY + OR | Intercrop: RY with 60 kg ha$^{-1}$ of seeds + OR with 10 kg ha$^{-1}$ of seeds | Early autumn (April) | December: velvet bean | At flowering, cover was bedded with a cutting roller |
| CT | Cover crop: pear millet (*Pennisetum glaucum* L.) | ---- | Pear millet | At flowering, cover was bedded with a cutting roller |
| NTS: millet (M) + velvet (V) + sunflower (S) | Intercrop: M (*Pennisetum glaucum* L.) + V (*Mucuna aterrima* Piper and Tracy) + S (*Helianthus annuus* L.) | ---- | Millet + velvet bean + sunflower | At flowering, the cover plants were killed with the herbicide glyphosate. |

NTVS—no-tillage vegetable system; CT—conventional tillage; NTS—no-tillage system. In the control treatment, the predominance of the species is described in Souza et al. [37].

The amounts of seeds per hectare were calculated based on the highest values recommended by Monegat [39] + 50% to ensure seed germination and formation of a higher dry mass during the onion cycle. Cover crops are winter species and were sown in early autumn (April) every year (Table 1). During the cover plant cycle, no fertilization, irrigations, or cultural practices were performed. It was not necessary to use irrigation, as there is no lack of rainfall in the region. Annual precipitation values are between 1400 and 2100 mm, according to Souza et al. [37].

Adjacent to the experiment, two other treatments were evaluated, one managed in the conventional tillage system (CT) and another managed in the no-tillage system (NTS), albeit not agroecological, using the herbicide glyphosate for desiccation of the straw for subsequent planting of the onion (Table 1). The treatment with a CT was evaluated alongside the no-tillage system, being the original onion crop area kept under CT for 20 years until 1996. From 2007, onion was grown in rotation with pear millet (*Pennisetum glaucum* L.) in the summer. At flowering, millet was bedded with a cutting roller, and, after 30–60 days, plowing was carried out, followed by grading for implantation of the onion crop. The soil in the 0–10 cm layer had 420 g kg$^{-1}$ of clay, 5.8 pH in water, 17.1 mg dm$^{-3}$ of available phosphorus, 80.0 mg dm$^{-3}$ of potassium exchangeable, 0.0 mmol$_c$ kg$^{-1}$ exchangeable aluminum, 7.3 cmolc kg$^{-1}$ exchangeable calcium, and 3.0 cmolc kg$^{-1}$ exchangeable magnesium [27]. Fertilization was performed according to regional recommendations [40]. The NTS was a succession of intercrops of soil cover crops (summer) and annual onion. The intercrops was millet + velvet bean (*Mucuna aterrima* Piper and Tracy) + sunflower

(*Helianthus annuus* L.). Approximately 14 days before onion planting, the plants (millet + velvet + sunflower) were killed with the herbicide glyphosate (360 g $L^{-1}$) at 4 L $ha^{-1}$. The NTS was implemented in April 2007, and in September 2016, nine years after the beginning of the experiment, soil samples were collected. The soil in the 0–10 cm layer had 326 g $kg^{-1}$ of clay; pH in water of 6.1; exchangeable Ca, Mg, Al of 6.4, 2.7, and 0.0 cmolc $dm^{-3}$ (extracted with KCl 1 mol $L^{-1}$), respectively; and available P and K of 42 and 208 mg $dm^{-3}$ (extracted with Mehlich-1), respectively. Fertilization was performed according to regional recommendations [40]. For comparative purposes, a forest area (secondary forest with ±36 years) adjacent to the experiment was collected and evaluated as soil condition without anthropic interference. The forest area presented, in 2013, 67.5 g $kg^{-1}$ of TOC, 116.37 of OM, and 4.0 g $kg^{-1}$ of N [27].

The experimental design was randomized blocks with four repetitions per treatment. Each experimental unit had 5 × 5 m, totaling 25 $m^2$. Every July, since the experiment implementation, all winter species were lodged using a cutting roller (Table 1). Then, they were applied to the area of 96 kg of $P_2O_5$ $ha^{-1}$ in the form of Gafsa rock phosphate, 175 kg of $P_2O_5$ $ha^{-1}$, 125 kg of $K_2O$ $ha^{-1}$, 100 kg of N $ha^{-1}$ in the form of chicken manure, half on planting the seedlings, and the remaining 45 days after. From the 2011 harvest, rock phosphate was not applied, as the levels were interpreted as very high [40]. The onion seedlings were produced in beds under the conventional tillage method, with the cultivar Empasc 352- Bola Precoce. The seedlings' transplant was carried out manually after opening furrows using an adapted no-tillage onion machine. The spacing was 0.40 m between rows and 0.10 m between plants, with ten rows of onions per plot. Manual weeding was performed in all treatments under the no-tillage vegetable system at 40 and 90 days after planting the onion seedlings to reduce the stand of spontaneous plants. In the conventional tillage system, spontaneous plants control was performed with herbicides. After the onion harvest in December of each year, velvet bean was sowed (*Mucuna aterrima* Piper and Tracy) in the treatments under the no-tillage vegetable system with 120 kg $ha^{-1}$ of seeds (Table 1), except in the first year (2009). Every April, the velvet bean was lodged, and, right after, winter cover plants were sowed. These procedures were repeated every year until the time of the soil sampling in 2019.

The average dry mass production (DM) of the cover plants in the experimental area in 2018 and 2019 is presented in Table 2. In VE, plants show slow initial growth and, at the end of their cycle, have low phytomass production compared to the planted cover plants and low nutrient cycling capacity [41,42].

The no-tillage vegetable system recommends the addition of more than 10 tons of DM $ha^{-1}$ per year [12]. In this experiment, the lower DM yields obtained are due to the early lodging of the cover species preceding full bloom, to later carry out the cultivar Empasc 352 Bola Precoce seedlings planting, which occurs in the second half of July, prevailing in the region where the present study occurred [43].

**Table 2.** Average dry matter production (Mg $ha^{-1}$) of the winter cover plants in the 2017/2018 and 2018/2019 crops in the onion crop no-tillage and conventional tillage systems.

| Dry Matter | Treatments | | | | | | | |
|---|---|---|---|---|---|---|---|---|
| | SV | RY | BO | OR | BO + OR | RY + OR | CT | NTS |
| | 2017/2018 | | | | | | | |
| Mg $ha^{-1}$ | --- | 4.5 | 4.6 | 4.0 | 4.2 | 4.7 | --- * | --- ** |
| | 2018/2019 | | | | | | | |
| Mg $ha^{-1}$ | 1.5 | 4.2 | 4.5 | 4.1 | 4.5 | 4.6 | --- * | --- ** |

SV—spontaneous vegetation; RY—Rye; BO—black oats; OR—oilseed radish; BO + OR—oats + oilseed radish; RY + OR—rye + oilseed radish; CT—conventional tillage system; NTS—no-tillage system (millet + velvet + sunflower). * According to Loss et al. [44], the millet dry matter production in CT grown in the summer reached an average of 12 Mg $ha^{-1}$. In addition, the seven-year velvet bean DM average production grown in the summer presented an average of 4.8 Mg $ha^{-1}$. ** According to Giumbelli et al. [26], the dry matter production in NTS grown in the summer reached an average of 11 Mg $ha^{-1}$.

### 2.2. Soil Sampling, Preparation, and Analysis

The soil sampling took place in April 2019 with the opening of 40 × 40 × 40 cm trenches between lines of each plot using a cutting shovel. Then disturbed samples were collected in the 0–5, 5–10, and 10–30 cm layers. The sampling was stored in plastic bags duly identified, the soil samples were air-dried, un-clod, and sieved in a 2 mm mesh to obtain a fine air-dried soil sample.

Undisturbed soil samples were collected at the same depths using the volumetric ring method to determine soil density (SD) [45]. The ring has a volume of 50 cm$^3$, and the soil in the rings is weighed after drying at 110 °C for 72 h. Then, the SD was obtained by dividing dry mass by the ring's volume, according to the Embrapa [45] methodology. The SD values were used to calculate the C and N stocks and are presented in Table 3. The weighted average diameter data are presented in Table 3, with the values available in full in Giumbelli et al. [26], Loss et al. [44], and Bortoloni et al. [5].

**Table 3.** Soil density and weighted average diameter under the forest, and no-tillage and conventional tillage system with different cover plants and the onion crop.

| Layers | Treatments | | | | | | | | |
|---|---|---|---|---|---|---|---|---|---|
| | **FOREST** | **SV** | **RY** | **BO** | **OR** | **BO + OR** | **RY + OR** | **CT** | **NTS** |
| | Soil Density (Mg m$^3$) | | | | | | | | |
| **0–5 cm** | 0.72 | 1.14 | 1.13 | 1.18 | 1.13 | 1.20 | 1.12 | 1.25 | 1.01 |
| **5–10 cm** | 0.72 | 1.35 | 1.34 | 1.32 | 1.34 | 1.25 | 1.29 | 1.28 | 1.08 |
| **10–30 cm** | 0.77 | 1.41 | 1.39 | 1.35 | 1.39 | 1.38 | 1.38 | 1.32 | 1.23 |
| | Weighted average diameter (mm) | | | | | | | | |
| **0–5 cm** | 4.62 | 4.80 | 4.84 | 4.85 | 4.60 | 4.74 | 4.76 | 2.43 | 4.60 |
| **5–10 cm** | 4.63 | 4.58 | 4.74 | 4.80 | 4.66 | 4.65 | 4.41 | 2.68 | 4.56 |

Forest—secondary forest; SV—spontaneous vegetation; RY—rye; BO—black oats; OR—oilseed radish; BO + OR—black oats + oilseed radish; RY + OR—rye + oilseed radish; CT—conventional tillage system; NTS—no-tillage system.

### 2.3. C and N Stocks Determination and Calculation

For the organic carbon determination (TOC) and total nitrogen (TN), TFSA was used and analyzed in an auto-analyzer at 900 °C (CHN—600 Carlo Erba EA—1110, Italy). The TOC and TN stocks were calculated using the equivalent weight method [46], according to the equation below:

$$C_S = \sum_{i=1}^{n-1} C_{Ti} + \left[ M_{Tn} - \left( \sum_{i=1}^{n} M_{Ti} - \sum_{i=1}^{n} M_{Si} \right) \right] C_{Tn}$$

where:

$C_S$ is the total stock in Mg C ha$^{-1}$;

$\sum_{i=1}^{n-1} C_{Ti}$ is the sum of the carbon/nitrogen of the first (surface) to the last layer in the soil profile in the evaluated treatment (Mg ha$^{-1}$);

$\sum_{i=1}^{n} M_{Ti}$ is the sum of the soil weight of the first to the last layer in the soil profile in the evaluated treatment (Mg ha$^{-1}$);

$\sum_{i=1}^{n} M_{Si}$ is the sum of the soil weight of the first to the last layer in the soil profile in the evaluated treatment (Mg ha$^{-1}$);

$M_{Tn}$ is the soil weight in the last layer of the soil profile in the evaluated treatment (Mg ha$^{-1}$);

$C_{Tn}$ is the carbon/nitrogen concentration in the last layer in the soil profile in the evaluated treatment (Mg C Mg$^{-1}$ soil).

The reference treatment was the forest (secondary forest), which presented the lowest SD values (Table 2), and consequently, the lowest equivalent weights per layer.

### 2.4. Particle-Size Fractioning of SOM

The particle-size fractioning was performed according to the methodology described by Cambardella and Elliott [25]. Initially, the soil samples were dispersed using 20 g of fine air-dried soil with 60 mL of sodium hexametaphosphate solution and stirred for 15 h in a horizontal shaker. Then, the material was passed through a 53 μm sieve to separate the sand fraction from the silt and clay fraction and dried at 50 °C. Subsequently, its weight was quantified and ground in porcelain mortar to obtain the content of particulate organic carbon (POC) and particulate organic nitrogen (PN) determined via elemental dry combustion analyzer (model Flashea 1112 Thermo Finnigan). The material that passed through the 53 μm sieve containing silt and clay minerals represents the mineral organic matter (MOM < 53 μm). To the MOM C and N contents (C-MOM and N-MOM), the difference between the total TOC/TN with POC/PN was measured. The equivalent weight method [46] was also used to calculate the C and N stocks of the particle-size fractions, as described above.

### 2.5. Statistical Analysis

The results were analyzed for normality and homogeneity through Lilliefors and Cochran tests, following the experimental design of randomized blocks with seven treatments (oats, rye, turnip, turnip + rye, turnip + oats, spontaneous vegetation, and the conventional tillage system) and four repetitions. The forest was used for comparison purposes, considered the closest condition to the soil without anthropic interference in the study region. The treatment results were submitted to variance analysis (ANOVA) with the application of the F test and the mean values, when relevant, compared to each other by the Scott–Knott test at 5% probability through the Assistat 7.7 Software.

## 3. Results

### 3.1. Total Soil Carbon and Nitrogen Stocks

In the surface layer of the soil (0–5 cm), the TOC (Figure 1) and TN (Figure 2) stocks were higher in the no-tillage vegetable system for the treatments with a cover plants intercrop (RY + OR and BO + OR) compared to single crops in the no-tillage vegetable system and also in the conventional tillage system treatments. In the 5–10 cm layer, OR and RY + OR treatments presented the highest stocks of TOC and TN. In the 10–30 cm layer, in the BO and BO + OR treatments, there are the highest stocks of TOC, while for the SV and RY treatments, there are the lowest stocks of TOC (Figure 1).

In the 0–30 cm layer, the treatments with single OR and in consortium with oats (BO + OR) and rye (RY + OR) stand out; they presented the higher stocks of TOC. The SV and RY treatments presented the lowest stocks of TOC. Regarding the conventional tillage system, the no-tillage vegetable system treatments with single- and intercropped BO and OR crops (BO + OR and RY + OR) showed higher stocks of TOC than the conventional tillage system in the 0–30 cm layer (Figure 1).

For TN stocks in the 10–30 cm layer, higher values were observed for the RY, BO, RY + OR, and conventional tillage system treatments, with a lower value found in SV treatment. The TN stock in the 0–30 cm layer was higher for RY + OR than other treatments (Figure 2). Regarding the reference area (forest), the TOC and TN stocks are proportionally higher than those found in the no-tillage vegetable system (Figures 1 and 2, respectively).

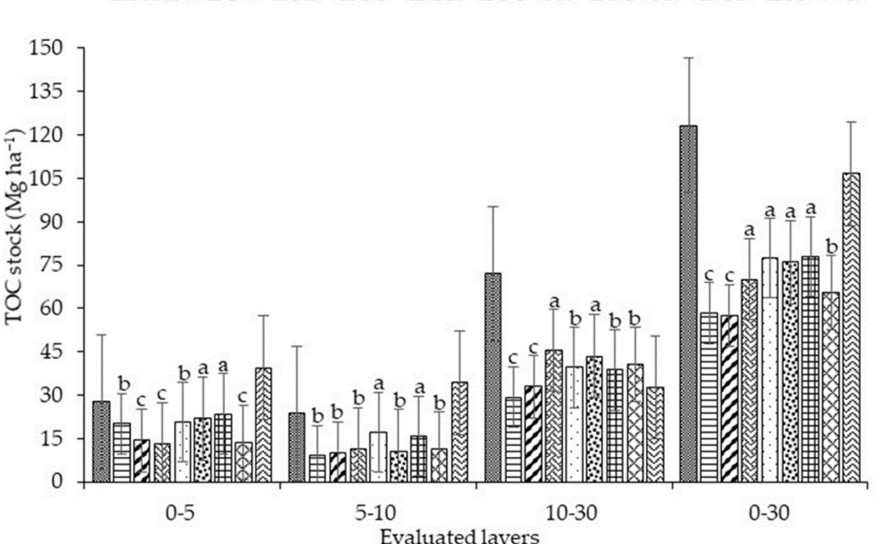

**Figure 1.** Total organic carbon (TOC) stocks in soil cultivated with onions in a non-tillage system with single and consortium cover plants and the conventional tillage system. Means followed by the same letter do not differ statistically by Scott–Knott test at 5% probability. Forest—secondary forest (reference area); SV—spontaneous vegetation; RY—rye; BO—black oats; OR—oilseed radish; BO + OR—black oats + oilseed radish; RY + OR—rye + oilseed radish; CT—conventional tillage system; M + V + S—millet + velvet + sunflower.

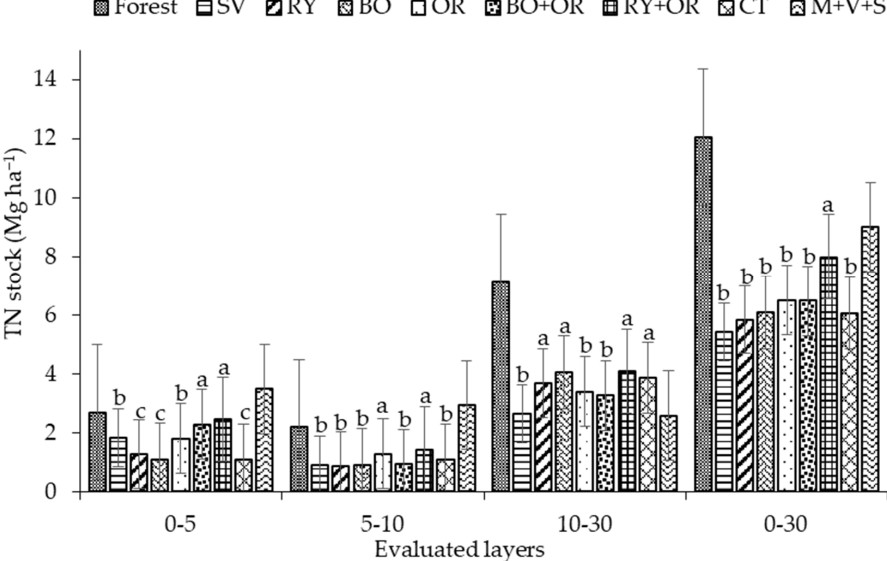

**Figure 2.** Total nitrogen stocks (TN) in soil cultivated with onions in a non-tillage system with single and consortium cover plants and the conventional tillage system. Means followed by the same letter do not differ statistically, by Scott–Knott test at 5% probability. Forest—secondary forest (reference area); SV—spontaneous vegetation; RY—rye; BO—black oats; OR—oilseed radish; BO + OR—black oats + oilseed radish; RY + OR—rye + oilseed radish; CT—conventional tillage system; M + V + S—millet + velvet + sunflower.

*3.2. C and N Stocks of the SOM Particle-Size Fractioning*

In the 0–5 cm layer, single RY and OR treatments showed the higher stocks of POC (Figure 3) and PN (Figure 4); the conventional tillage system treatment showed the lowest stocks of POC and PN. In the 5–10 cm layer, single RY treatments and intercropped RY + OR stand out with higher POC and PN stocks. It is also noteworthy that the conventional

tillage system showed the lowest values regarding PN stocks (Figure 4). In general, in the 10–30 and the 0–30 cm layers, the RY and OR treatments presented the higher stocks of POC and PN, emphasizing the conventional tillage system, which, in the 0–30 cm layer, showed the lowest values.

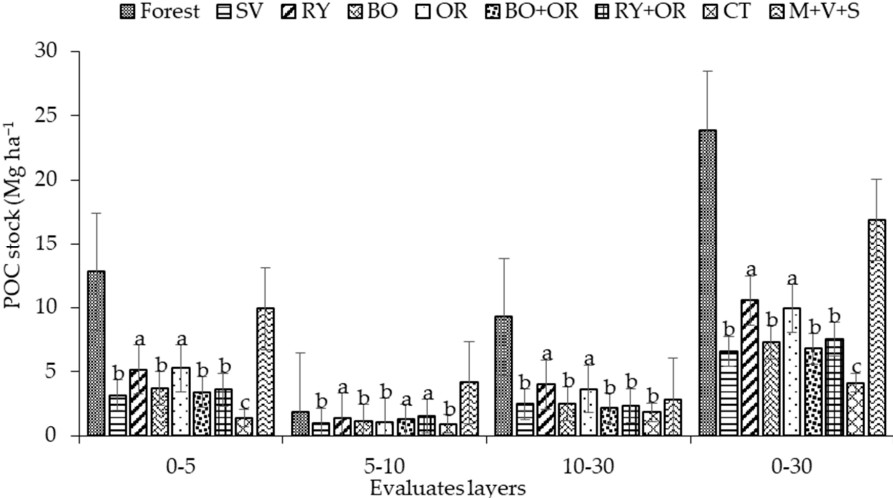

**Figure 3.** Particulate organic carbon stocks (POC) of the particle-size fraction of the soil's organic matter cropped with onions in a no-tillage system with single and intercropped cover plants and the conventional tillage system. Means followed by the same letter do not differ statistically, by Scott–Knott test at 5% probability. Forest—secondary forest (reference area); SV—spontaneous vegetation; RY—rye; BO—black oats; OR—oilseed radish; BO + OR—black oats + oilseed radish; RY + OR—rye + oilseed radish; CT—conventional tillage system; M + V + S—millet + velvet + sunflower.

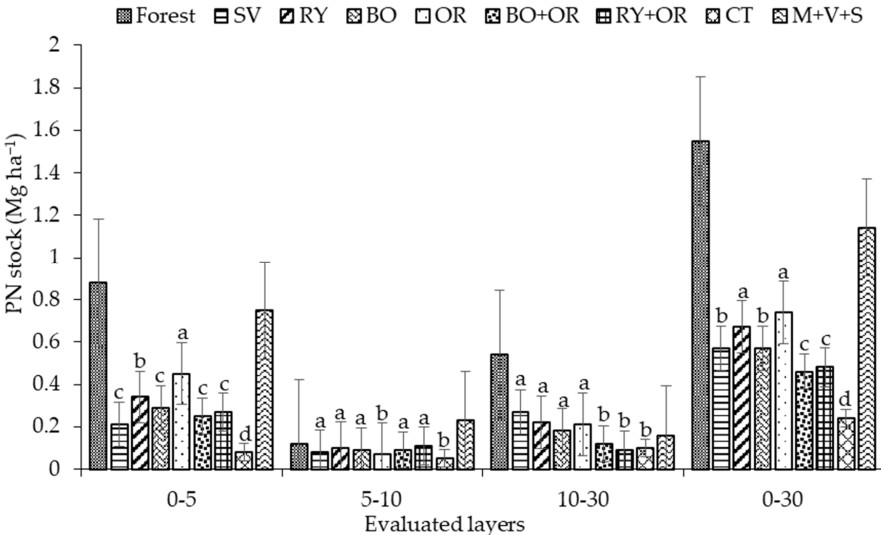

**Figure 4.** Particulate nitrogen stocks (PN) of the particle-size fraction of the soil's organic matter cropped with onions in a no-tillage system with single and intercropped cover plants and the conventional tillage system. Means followed by the same letter do not differ statistically, by Scott–Knott test at 5% probability. Forest—secondary forest (reference area); SV—spontaneous vegetation; RY—rye; BO—black oats; OR—oilseed radish; BO + OR—black oats + oilseed radish; RY + OR—rye + oilseed radish; CT—conventional tillage system; M + V + S—millet + velvet + sunflower.

The higher stocks of MAOC and MN In the 0–5 cm layer occurred in the treatment RY + OR, followed by SV and BO + OR (Figures 5 and 6). In the 5–10 cm layer, the higher stocks were evidenced in the treatment with single OR, followed by RY + OR and the conventional

tillage system. In the 10–30 cm layer, the lowest stocks were found in the SV. While in the 0–30 cm layer, the intercrop of BO + OR and RY + OR showed the highest values of MAOC when compared to single treatments in the no-tillage vegetable system (Figure 5), and MN when compared to all treatments (Figure 6).

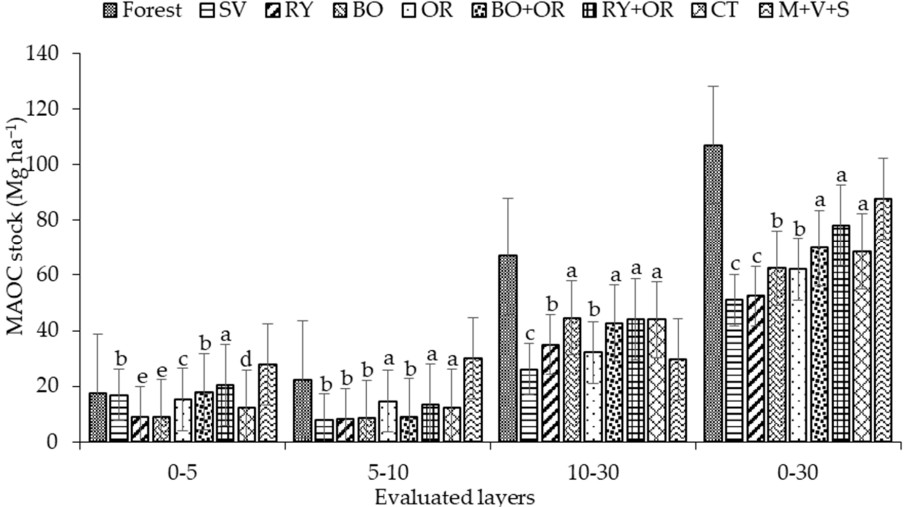

**Figure 5.** Mineral-associated organic carbon stocks (MAOC) of the particle-size fraction of the soil's organic matter cropped with onions in a no-tillage system with single and intercropped cover plants and the conventional tillage system. Means followed by the same letter do not differ statistically, by Scott–Knott test at 5% probability. Forest—secondary forest (reference area); SV—spontaneous vegetation; RY—rye; BO—black oats; OR—oilseed radish; BO + OR—black oats + oilseed radish; RY + OR—rye + oilseed radish; CT—conventional tillage system; M + V + S—millet + velvet + sunflower.

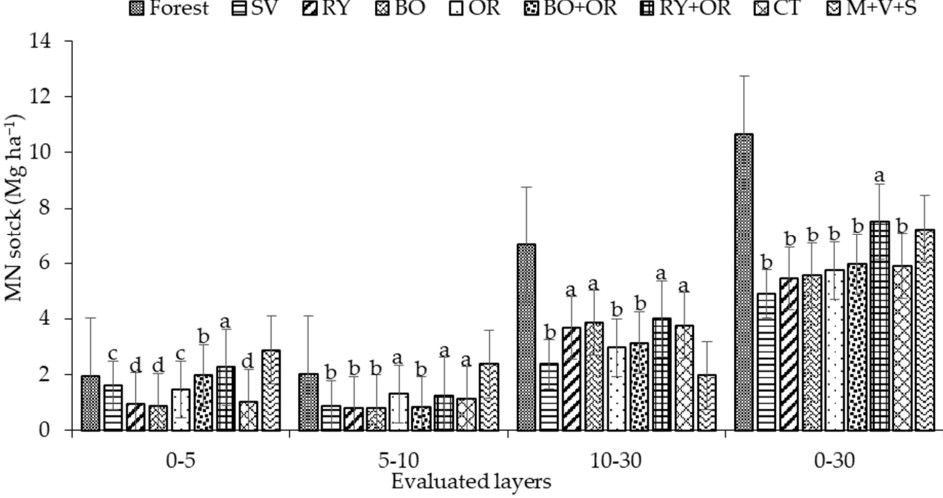

**Figure 6.** Mineral-associated nitrogen stocks (MN) of the particle-size fraction of the soil's organic matter cropped with onions in a no-tillage system with single and intercropped cover plants and the conventional tillage system. Means followed by the same letter do not differ statistically, by Scott–Knott test at 5% probability. Forest—secondary forest (reference area); SV—spontaneous vegetation; RY—rye; BO—black oats; OR—oilseed radish; BO + OR—black oats + oilseed radish; RY + OR—rye + oilseed radish; CT—conventional tillage system; M + V + S—millet + velvet + sunflower.

## 4. Discussion

High stocks of TOC and TN in forest areas are expected because they are systems without anthropic interference. These stocks' maintenance in the soil occurs through plant

activity, which, with the constant burlap deposition, makes the environment conducive to biota activity [47–49]. In the treatment M + V + S, the high values of TOC and TN stocks are due to the addition of dry matter (11 Mg ha$^{-1}$), associated with the non-revolving of the soil (NTS) and the use of three plant species from different botanical families [6,26]. This pattern was not observed in the CT treatment, because the soil is turned over in the CT, which fragments the millet straw, increasing its decomposition, with subsequent losses of C and N [44].

In the no-tillage vegetable system treatments, the higher TOC stocks in the surface layer (0–5 cm) in the RY + OR and BO + OR treatments are due to the rapid growth of single and intercropped OR, producing a large amount of biomass and adding organic matter to the soil and, consequently, adding carbon. This mechanism was observed by Doneda et al. [50], who found higher DM yields in the grass intercrop with forage turnip in their research. The result is due to the rapid initial growth of this cover plant. This OR effect can also be found on the 0–30 cm layer.

The replacement of the forest area vegetation by crops that use soil mobilization, i.e., the conventional tillage system, causes the reduction of soil's TOC and TN stocks, mainly in the surface layer of the soil [51,52]. Moreover, this reduction pattern can be observed in this study, emphasizing the 0–5 cm layer in which the conventional tillage system presented the lowest values of TOC and TN stocks, albeit not differing from single treatments (BO and RY). These results are due to the plowing and grading practices carried out in the conventional tillage system, causing soil aggregates to break down, with a consequent increase in the soil organic matter decomposition and mineralization previously protected inside the aggregates [53]. The similarity of C and N stock results in the conventional tillage system compared to the no-tillage vegetable system, mainly at greater depths, is due to the summer millet crop in the conventional tillage system area. The millet DM production ranges from 10 to 12 Mg ha$^{-1}$ [44]; in addition to its high C/N ratio and root system intense activity, it still generates C and N values similar to the no-tillage vegetable system, even in the conventional tillage system.

Soil use and management systems with reduced mobilization or no turnover, such as eucalyptus reforestation, pasture, and no-tillage vegetable system, can recover C and N stocks compared to native vegetation areas, even reaching higher stocks than those of native vegetation [54]. Corazza et al. (1999) concluded that the conventional tillage system acts as a source of atmospheric carbon and the no-tillage vegetable system as a carbon deposit in the soil. The data generated on this study corroborates the previous statements regarding C and N stocks. The no-tillage vegetable system with intercropped cover plants (RY + OR) exceeded the C and N stocks in the analyzed soil profile (0–30 cm) compared to the conventional tillage system and the no-tillage vegetable system with spontaneous vegetation and other treatments with cover plants. Several studies in the literature restate the benefits of using the no-tillage vegetable system regarding C and N stocks [55–59]. Notably, for the C stocks in the 0–30 cm layer, the cover plants intercrop (RY + OR and BO + OR) was more efficient in increasing the C stocks than RY, BO, and SV single crops.

OR has a sturdy root system, capable of reaching high depths, cycling nutrients from the deeper layers of the soil, and adding high amounts of plant material to the soil surface [60]. Thus, single- or intercropped OR favored the TOC accumulation in the 0–30 cm layer (Figure 1) and the accumulation of N when the OR was intercropped with RY (Figure 2). In a study developed by Bayer et al. [61], the no-tillage vegetable system areas promoted a 9% increase in the TOC stock compared to the conventional tillage system; this effect was restricted to the 0–20 cm layer and dependent on the crop system.

The higher TN stocks in treatments with OR are probably due to the accelerated development of this crucifer and the easy decomposition of its biomass [42]. Unlike grasses in single treatments, which have a high C/N ratio, high levels of remaining lignin, high lignin/N ratio, and low cellulose/lignin ratio, structures with slow and gradual decomposition and mineralization are resistant to microbial decomposition [62]. The intercrop between rye + oilseed radish and black oats + oilseed radish accumulate higher



amounts of N than their single crops, proving the performance of turnip as a plant capable of cycling N, even despite it not being a legume [50].

Cover plants in the no-tillage vegetable system are responsible for entering, cycling, and storing TN into the soil. In addition, the soil mobilization restricted to the planting row culminates in the soil organic matter preservation and N accumulation [63]. The amount of accumulated TN is influenced by the cover plants and soil tillage systems. After nine years of experiment, the authors found that the amount of TN accumulated in the surface layer occurred mainly in systems with minimal soil mobilization. Systems with reduced mobilization in soil tillage, such as the no-tillage vegetable system, have positive effects on the N accumulation in the surface layers of the soil, being an ally for biological activities and the reduction of environmental risks [64].

The lower stocks of POC and PN in the conventional tillage system treatment, compared to the no-tillage vegetable system, are justified by soil management. Due to the periodic mobilization of the soil to prepare the area, there is a breakdown of soil aggregates that causes the C and N exposure. These aggregates were physically protected against microbial activity, increasing their decomposition, with consequent reduction of C and N content, as evidenced [21,44,53]. Soil disturbance with plowing, grading, and rotary hoe practices breaks macroaggregates. Drying–moistening of the soil intensifies this aggregate disruption [65]. Therefore, microaggregates, physically protected by macroaggregates, are exposed to biodegradation; thus losses of C and N occur.

On the other hand, higher stocks of POC and PN in the no-tillage vegetable system areas are related to the maintenance of plant residues on the soil surface and the absence of soil mobilization. Thus, the no-tillage vegetable system favors the physical protection of aggregates (mainly macroaggregates), protecting the labile fraction of soil organic matter (POC and PN). The soil under the no-tillage vegetable system must be continuously maintained with plant residues so that efflux of C and N from the labile fraction of soil organic matter into the atmosphere does not occur [61].

The use of the no-tillage vegetable system on the POC and TOC stocks compared to the conventional tillage system led to POC accumulation in the 0–20 cm layer due to the greater annual addition of C through plant residues, which, kept on the soil surface, decrease the organic material microbial decomposition rate [66]. POC was more sensitive to soil management than TOC stock. Moreover, the most significant changes in C stocks occurred in the soil surface layer, 0–5 cm, where POC was 58% higher in the no-tillage vegetable system than in the conventional tillage system, and TOC, only 32%. The POC acts as a source of energy for microorganisms, resulting in the greater stability of aggregates, and mainly macroaggregates. The TOC works in the more humified fraction of the soil organic matter, acting on microaggregate stability. The authors concluded that the no-tillage vegetable system triggers processes that increase the stability of aggregates and stocks of soil organic matter.

As for POC stocks, PN stocks are higher in areas where plant residues cover the soil surface. Regarding the POC and PN stocks in the forest area, the no-tillage vegetable system has values proportionally closer to the forest area than the conventional tillage system, indicating that soil management in the no-tillage vegetable system is more efficient in adding phytomass to the soil and accumulating C and N. These results are corroborated by Nunes et al. [63], who evaluated the effects of the conventional tillage system with millet, the no-tillage vegetable system with millet, and the no-tillage vegetable system with velvet bean regarding C and N stocks in the particle-size fractions of SOM. PN stock values were higher in the no-tillage vegetable system for both coverage systems than in the conventional tillage system. According to the authors, due to the greater sensitivity of the particulate fraction to management systems, this fraction is more dependent on structural protection through soil aggregation. This means that the conventional tillage system, by using implements such as a disk plow and disc harrow, promotes the fragmentation of surface plant material, and the aggregates break down, resulting in the exposure of the soil organic matter to decomposition by microorganisms. The PN gains observed in the

no-tillage vegetable system with millet and velvet bean were 50.7 and 34.9%, respectively, compared to the conventional tillage system [63].

The MAOC and MN stocks values in the intercrop of RY + OR were proportionally higher than the values of the forest area in the surface layer of the soil (about 14%) and higher than the conventional tillage system. These results indicate that the intercrop of cover plants with RY + OR has the potential to recover the more humified and more stable fraction of the soil organic matter since the replacement of the conventional tillage system by the no-tillage vegetable system with the intercrop of grasses (RY) and cruciferous (OR) recovered and raised the stocks of MAOC and MN after 11 years of experiment. These results also corroborate the higher stocks of TOC (Figure 1) and TN (Figure 2) found in the RY + OR intercrop. Due to higher C input and improved soil quality via long-term conservationist management, the no-tillage vegetable system is higher than the conventional tillage system with respect to carbon sequestration rates. The particulate fraction (POC/PN) is an indicator of soil management efficiency, and the MAOC/MN is a vital drain and reservoir of atmospheric C and N. The adding of cover plants is a fundamental strategy for the storage of C and N in the soil [67].

In the 0–30 cm layer, the SV treatment presented the lowest stocks of MAOC and MN due to the low phytomass production of spontaneous plants and the lower contribution of organic matter to the soil, culminating in lower stocks of C and N of the most stable fractions, as well as lower TOC and TN stocks (Figures 1 and 2). In this same layer, the intercrop RY + OR treatment presented higher MAOC and MN stocks, to the detriment of the single cover plants crop and the conventional tillage system, corroborating the TOC and TN stocks. The conventional tillage system treatment showed high stocks of MAOC and MN in the 0–30 cm layer because the soil management in this system reduces the labile fraction (POC and PN), and due to the millet crop in summer that produces high C/N ratio biomass, the most stable fractions, MAOC and MN, remain in the soil in higher proportions [6,53].

Treatments with single- or intercropped cover crops in the no-tillage vegetable system did not differ from each other in terms of soil aggregation in the surface layer, with weighted average diameter (WMD) values ranging from 4.60 to 4.85 mm, and did not differ from the forest area, which presented a WMD of 4.62 mm [68]. The M + V + S treatment presented a WMD of 4.60 mm [26]. However, all were superior ($p < 0.05$) to treatment with a conventional tillage system, which presented a WMD of 2.54 mm [44]. No-tillage vegetable system treatments also showed higher values of total glomalin (TG) and easily extractable glomalin (EEG) compared to the conventional tillage system, which showed values ($p < 0.05$) of 125.88 and 34.03 $\mu g\ g^{-1}$ of TG and EEG, respectively. The treatments in the no-tillage vegetable system, which did not differ among themselves, showed values ranging from 209.43 to 245.39 $\mu g\ g^{-1}$ and 54.24 to 65.91 $\mu g\ g^{-1}$ EEG. It is noteworthy that the forest area presented values of GT and EEG equal to those of treatments in the no-tillage vegetable system, with values of 234.21 and 64.12 $\mu g\ g^{-1}$ of TG and EEG, respectively [68].

The higher TOC and TN stocks and the particle-size fractions of SOM in the treatments in the agroecological no-tillage system and in the no-tillage system with millet + velvet + sunflower corroborate the higher rates of soil aggregation compared to the conventional tillage system. The use of millet in the no-tillage system treatment, associated with the absence of soil disturbance, favors higher C and N stocks in the soil compared to the use of millet in the conventional tillage system.

The six-year evaluation of the no-tillage vegetable system effect in four autumn crop systems in rotation with commercial soybean–corn plants, compared to the conventional tillage system with summer crops, on the impact on carbon storage in particulate fractions indicated that MAOC stocks in the 0–20 cm layer were statistically equal for the no-tillage vegetable system management system, justified via the short-term period. The slightest change in the C and N fraction related to minerals (MAOC and MN) is due to this fraction's advanced stage of humification and stability, being located inside stable microaggregates, and its higher chemical persistence. A more extended period is required for management

practices to generate effects on MAOC stocks. The conventional tillage system presented the lowest MAOC stocks compared to the no-tillage system, indicating that the system causes the aggregates to break down and soil organic matter losses [61].

The M + V + S treatment showed higher POC and PN stocks compared to the other treatments (0–5 and 0–30 cm). These results may be due to the large amount of biomass produced, in addition to having plants from the grass and leguminous families. However, the RY + OR treatment stood out with, higher MN values in the 10–30 and 0–30 cm layer compared to the M + V + S.

## 5. Conclusions

The no-tillage vegetable system has the higher TOC and TN stocks, using grasses and brassica intercropped to the detriment of single crops, and it is superior to the no-tillage vegetable system with spontaneous vegetation and the conventional tillage system.

The no-tillage vegetable system increases the C and N stocks of the soil organic matter's more labile fraction of POC and PN than the conventional tillage system. The POC and PN stocks were favored in the areas with single RY and OR. The consortium of millet + velvet + sunflower in the no-tillage system showed higher TOC, TN, POC, and PN stocks compared to the other treatments (0–5 and 0–30 cm).

The RY + OR intercrop increased the C and N of the soil organic matter in more stable fractions (MAOC and MN) in the 0–30 cm layer, reflecting the high TOC and TN stocks in this layer. In general, the intercrop of cover plants is ideal for obtaining the no-tillage vegetable system's maximum potential, favoring several mechanisms between soil, plant, and atmosphere, and resulting in improved soil quality, increased organic matter, and higher stocks of C and N.

**Author Contributions:** Conceptualization, A.L., C.R.L., C.K. and J.J.C.; methodology, A.C.K., A.P.L., T.S.d.S. and L.D.G.; software, A.C.K. and A.L.; validation, A.C.K., G.B. and B.S.V.; formal analysis, A.C.K.; investigation, A.L., J.J.C.; resources, M.d.C.P. and J.J.C.; data curation, A.C.K.; writing—original draft preparation, A.C.K.; writing—review and editing, A.L., J.J.C., J.L.R.T. and G.B.; visualization, M.d.C.P. and C.R.L.; supervision, J.J.C. and A.L.; project administration, C.K. and J.J.C.; funding acquisition, J.J.C., A.L. and M.d.C.P. All authors have read and agreed to the published version of the manuscript.

**Funding:** We are grateful to the Programa de Bolsas Universitárias de Santa Catarina (UNIEDU) for granting a scholarship FUMDES, and for the financial support of the National Council for Scientific and Technological Development (CNPq), process nos. 405026/2021-8 (Universal) and 311474/2021-7 (PQ Scholarship), and the Coordenação de Aperfeiçoamento de Pessoal de Nível Superior (Capes), process n. 88887.691713/2022-00.

**Data Availability Statement:** Data available in a publicly accessible repository that does not issue DOIs. Publicly available datasets were analyzed in this study. This data can be found here: [https://repositorio.ufsc.br/bitstream/handle/123456789/227016/PAGR0474-D.pdf?sequence=-1].

**Conflicts of Interest:** The authors declare no conflict of interest.

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
