# Peer review of "Effects of Tillage and Cover Crops on Total Carbon and Nitrogen Stocks and Particle-Size Fractions of Soil Organic Matter under Onion Crop"

_horticulturae, doi:10.3390/horticulturae9070822_

Round 1

Reviewer 1 Report

The paper examines the long-term effects of tillage systems on onion crop, specifically focusing on the total carbon and nitrogen stocks, as well as particle-size fractions of soil organic matter. The manuscript is well-written in proper English and format. The experimental design and statistical approach appear to be adequate. However, there are several areas that require clarification and justification before considering this paper for publication.

  1. The authors did not provide any information about the initial soil carbon, nitrogen, and organic matter content for their experiment conducted in 2009, or for the forest area studied in 1996.
  2. The authors did not conduct an analysis of the mentioned parameters (e.g., root, leaf) in the onion crop to compare with the soil. While it may be beyond the scope of this study, it should be explicitly mentioned and acknowledged.
  3. The authors state that no irrigation was performed during the experiment, assuming that the onion crop's water requirements were met solely through rainfall. However, it is important to note that rainfall amounts can vary across different years, which can have a significant impact, particularly on the soil nitrogen content, which is soluble in water. This point should be discussed in light of the obtained results.
  4. The authors did not provide any information about the distribution of soil particle sizes. According to the results, it is evident that the soil can be divided into two layers: 0-10 cm and 10-30 cm. It would be beneficial to include this information to enhance the clarity of the study.
  5. In the abstract, the authors should specify what 'CT' stands for in line 4.

The manuscript is well-written in proper English and format.

Author Response

Reviewer 1.

1-The authors did not provide any information about the initial soil carbon, nitrogen, and organic matter content for their experiment conducted in 2009, or for the forest area studied in 1996.

Authors: The organic carbon content is already informed in the manuscript (at the end of the first paragraph of Material and Methods). The other information (organic matter, total nitrogen) was inserted. For the forest area, the TOC, OM and N contents are from the year 2013.

  1. The authors did not conduct an analysis of the mentioned parameters (e.g., root, leaf) in the onion crop to compare with the soil. While it may be beyond the scope of this study, it should be explicitly mentioned and acknowledged.

Authors: Unfortunately we don't have that information. As mentioned by the reviewer, these analyzes were not the objective of the study.

  1. The authors state that no irrigation was performed during the experiment, assuming that the onion crop's water requirements were met solely through rainfall. However, it is important to note that rainfall amounts can vary across different years, which can have a significant impact, particularly on the soil nitrogen content, which is soluble in water. This point should be discussed in light of the obtained results.

Authors: It was not necessary to use irrigation, as there is no lack of rainfall in the region. Annual precipitation values are between 1400 and 2100 mm, according to Souza et al. (2021).

  1. The authors did not provide any information about the distribution of soil particle sizes. According to the results, it is evident that the soil can be divided into two layers: 0-10 cm and 10-30 cm. It would be beneficial to include this information to enhance the clarity of the study

Autohors: In the material and methods there is information that the soil has 380 g kg of clay. However, we also entered the amounts of sand (420 g kg-1) and silt (200 g kg-1), as well as the textural class (clay loam)

In the abstract, the authors should specify what 'CT' stands for in line 4.

Authors: Added (Conventional tillage)

Reviewer 2 Report

The article consisted of six no-tillage single and intercropping treatments, compared to conventional tillage, and evaluated the stocks of total organic carbon (TOC), total nitrogen (TN), particulate organic carbon (POC), particulate nitrogen (PN), mineral-associated organic carbon (MAOC), and mineral-associated nitrogen (MN) in the 30 cm soil layer after long-term operation. The research results have significant implications for the sustainable utilization of soil. However, the article has the following issues:

  1. In the abstract, it is mentioned that "MAOC and MN stocks were higher than forest in RY+OR intercrop," but this result only applies to the 0-5 cm soil layer. In the 0-30 cm soil layer, the results are actually opposite, and this point needs to be corrected.
  2. What is the scientific basis for sampling the soil at 0-5 cm, 5-10 cm, and 10-30 cm? Why not sample at every 5 cm interval?
  3. There are inconsistencies between the treatment names in the figures and the textual description. For example, VE in the figures should be SV, FT should be OR, and OA+FT2 in Figures 4 and 5 should be RY+OR, etc.
  4. The English translation in the article requires polishing or modification to align with English expression conventions.
  1. The English translation in the article requires polishing or modification to align with English expression conventions.

Author Response

In the abstract, it is mentioned that "MAOC and MN stocks were higher than forest in RY+OR intercrop," but this result only applies to the 0-5 cm soil layer. In the 0-30 cm soil layer, the results are actually opposite, and this point needs to be corrected.

Authors: We have corrected the information in the summary (only to topsoil layer). And  in the 0-30 cm layer, the intercrop of BO+OR and RY+OR showed the highest values of MAOC when compared to single treatments in the vegetable no-tillage system (Figure 5), and MN when compared to all treatments (Figure 6).

What is the scientific basis for sampling the soil at 0-5 cm, 5-10 cm, and 10-30 cm? Why not sample at every 5 cm interval?

Authors: In the no-tillage system, the highest inputs of organic matter occur in the first 10 cm of soil. And in the conventional tillage system, the greatest losses occur in the first 10 cm. For this reason sampling of 0-5 and 5-10 cm was used. For the assessment of C and N stocks, the IPCC recommends 0-30 cm; therefore, one more layer (10-30 cm) was evaluated, in order to have C and N stocks up to 30 cm (0-30 cm).

There are inconsistencies between the treatment names in the figures and the textual description. For example, VE in the figures should be SV, FT should be OR, and OA+FT2 in Figures 4 and 5 should be RY+OR, etc

Authors: All figures have been corrected.

The English translation in the article requires polishing or modification to align with English expression conventions.

Authors: A reading was done by a native speaker in the English language. See the modifications throughout the text.

Reviewer 3 Report

The high hydroponic inputs of vegetable cultivation are a potential threat to soil health, and therefore sound tillage is important for the sustainable use of vegetable farmland. This paper analyses changes in soil organic carbon from long-term no-till onion cultivation and obtains interesting results. However the objectives of the study were not clear and the data analysis was not comprehensive and systematic enough.

The research topic was the effect of tillage on organic matter, however, the two tillage treatments in the experiment were under very different cover or intercropping crops, so it was not possible to compare them and to distinguish between the effects of tillage or cover crop.

Secondly, the data were too singular and not analysed in detail. There is only organic matter data in the paper, but no explanation of why intercropping increases mineral bound carbon, and no relevant data such as soil aggregates or microorganisms to support this.

Therefore, it is recommended that the article redefines the objectives of the study and adds relevant data in order to obtain supporting conclusions.

  Figure 1. Pleas check the accuracy of the 0-30 cm data, the markings of significant differences appear to be inconsistent.

Author Response

The high hydroponic inputs of vegetable cultivation are a potential threat to soil health, and therefore sound tillage is important for the sustainable use of vegetable farmland. This paper analyses changes in soil organic carbon from long-term no-till onion cultivation and obtains interesting results. However the objectives of the study were not clear and the data analysis was not comprehensive and systematic enough.

The research topic was the effect of tillage on organic matter, however, the two tillage treatments in the experiment were under very different cover or intercropping crops, so it was not possible to compare them and to distinguish between the effects of tillage or cover crop.

Secondly, the data were too singular and not analysed in detail. There is only organic matter data in the paper, but no explanation of why intercropping increases mineral bound carbon, and no relevant data such as soil aggregates or microorganisms to support this.

Therefore, it is recommended that the article redefines the objectives of the study and adds relevant data in order to obtain supporting conclusions.

Authors: Dear reviewer, we disagree with the comment, as our objective was exactly to compare the soil management system (conventional tillage system versus no-tillage system for vegetables), as farmers in our region use these two systems with different types of cover crops (single or intercropped). And in this same experiment, we have consistent (published) data that treatments that are increasing C and N stocks are also increasing soil aggregation and soil microbiological activity. Information will be added to the text to reinforce the chemical benefits, along with the physical and biological ones.

Treatments with single or intercropped cover crops in the vegetable no-tillage system did not differ from each other in terms of soil aggregation in the surface layer, with weighted average diameter (WMD) values ranging from 4.60 to 4.85 mm, and also did not differ from the forest area, which presented a DMP of 4.62 mm. However, all were superior (p<0.05) to treatment with a conventional preparation system, which presented a DMP of 4.46 mm. SPDH treatments also showed higher values of total glomalin (GT) and easily extractable glomalin (GFE) compared to SPC, which showed values (p<0.05) of 125.88 and 34.03 μg g-1 of GT and GFE , they respected you. The treatments in SPDH, which did not differ among themselves, showed values ranging from 209.43 to 245.39 μg g-1 GT and 54.24 to 65.91 μg g-1 GFE. It is noteworthy that the forest area presented values of GT and GFE equal to those of treatments in SPDH, with values of 234.21 and 64.12 μg g-1 of GT and GFE, respectively.

Figure 1. Pleas check the accuracy of the 0-30 cm data, the markings of significant differences appear to be inconsistent.

Authors: There was an error in the woods area error bar. Got corrected. The other treatments were all correct, with no statistical errors.

Round 2

Reviewer 3 Report

Admittedly, it is interesting to compare carbon and nitrogen changes in different soil management systems. However the tillage practices in the title of this study do not cover the full range of soil management. This paper is supposed to compare the synergistic effects of tillage and cover crops. If a treatment of a millet cover in no-till is added, the results will be more meaningful, and the respective effects of tillage and cover on carbon storage can be effectively distinguished.

  In the revised text, a discussion of agglomerates was added, but no corresponding data were seen in the results section, nor was the relationship with carbon stocks analysed.

Therefore, it is recommended that the authors re-submit the paper with a new title and additional data.

Author Response

Dear Editor,
Reviewer requests have been granted.
We changed the title. And we inserted data from a treatment in a no-tillage system using millet+velvet+sunflower consortium for comparison purposes with the conventional tillage system treatment.
We also entered soil aggregation data in Table 2.

Best regards,

Round 3

Reviewer 3 Report

  The authors have made noted changes and added data in response to the revisions. Minor revisions are suggested for publication.

   The experimental treatments in the text involve more information on tillage, intercropping and mulching. For better understanding by the reader, it is suggested to add a table clearly indicating the name of the intercrop or cover crop, sowing and harvesting (disposal) time for each treatment.

Author Response

Dear editor,
I send the manuscript with Table 1 inserted in the text, according to the reviewer's recommendations.
We also did a reading of the manuscript to see repeated words. But, what is repeated the most is the description of the experiment, which, as it is a long-term experiment, we have already published several articles. And this description is the same. This cannot be considered plagiarism.
